# The Experiences of Newly Graduated Nurses during Their First Year of Practice

**DOI:** 10.3390/healthcare11142048

**Published:** 2023-07-17

**Authors:** Hanan F. Alharbi, Jamila Alzahrani, Amira Hamed, Abdulslam Althagafi, Ahmed S. Alkarani

**Affiliations:** 1Maternity and Child Health Nursing Department, College of Nursing, Princess Nourah bint Abdulrahman University, Riyadh 11671, Saudi Arabia; hfalharbi@pnu.edu.sa; 2Ministry of Health, Taif 26523, Saudi Arabia; 3Children’s Hospital, Ministry of Health, Taif 76200, Saudi Arabia; yassmena07@gmail.com (A.H.); abusalama@moh.gov.sa (A.A.); 4Department of Nursing, Applied Medical Sciences College, Taif University, Al Mathnah, Taif 20001, Saudi Arabia

**Keywords:** newly graduated nurses, transition, workplace, influencing factors, Saudi Arabia

## Abstract

The present study aimed to explore the experiences of newly graduated nurses during their first year of practise. A qualitative descriptive design was employed in this study. In-depth, semi-structured interviews were conducted with newly graduated nurses to gather detailed descriptions and experiences during their transition to the workplace in the first year after graduation. Thematic analysis was utilised to identify patterns and themes in the collected data. Ethical considerations were strictly enforced throughout the study. There are two main themes: factors contributing to the integration of new nurses into the workplace and the difficulties faced by new nurses in a work environment. Within the first theme, three subthemes emerged: the positive role of trainers, the gradual handling of patients, and the benefit of pre-employment training and volunteering. The theme of difficulties faced included three subthemes: difficulty dealing with the health system and devices, fear of dealing with new patients, and difficulty applying policies and procedures in the workplace. The study provides insights into the challenges faced by newly graduated nurses and the factors that contribute to their integration into practise settings. Educational departments in hospitals’ support and efficient access to policies are crucial for these nurses as they begin their early professional years.

## 1. Introduction

The transition from being a nursing student to becoming a professional nurse marks a significant milestone in the nursing profession [1,2]. This transition is accompanied by various struggles, such as being tense and suffering physical exhaustion, that newly graduated nurses (NGNs) must adapt to and cope with. It signifies the culmination of years of education and training and marks the beginning of their professional practise. However, this transition is not without its obstacles [1,3,4,5,6].

NGNs face several difficulties as they navigate the unfamiliar clinical environment and work culture. These difficulties can impede the integration and functioning of NGNs within the healthcare team. The challenges faced by NGNs during this transition period can be multifaceted and demanding [3,7].

Firstly, they must adapt to the new dynamics and demands of the clinical environment. The shift from the controlled and structured setting of the classroom to the fast-paced, high-pressure reality of healthcare settings can be overwhelming. NGNs are suddenly faced with the responsibility of caring for real patients, making critical decisions, and managing complex healthcare scenarios. This transition requires them to adjust quickly and apply their theoretical knowledge and skills to real-world situations [8,9]. In addition to the clinical challenges, NGNs also need to acclimatise to the work culture within healthcare settings. Each healthcare facility has its own unique organisational culture, norms, and expectations. NGNs must familiarise themselves with the specific policies and procedures of their workplace, understand the dynamics of interprofessional teams, and learn to collaborate effectively with colleagues from various disciplines. Building relationships and establishing effective communication channels are crucial aspects of integrating into the work environment [3,10]. Furthermore, NGNs may experience a lack of confidence in their abilities during this transition period. The shift from the supportive environment of a student to the autonomous role of a professional nurse can be daunting. NGNs may question their clinical skills, decision-making capabilities, and overall competence [11]. Moreover, NGNs often encounter challenges in managing their patient workload effectively. Balancing multiple patients’ needs, prioritising care tasks, and ensuring timely and safe interventions require efficient time management and organisational skills. NGNs may initially struggle with these aspects, leading to stress and feeling overwhelmed [10,12]. Additionally, workload may cause nurses to stop working on the clinical side [13]. Consequently, it is essential to understand the factors that contribute to the successful integration of NGNs into the work environment [14] and to understand Benner’s model and Duchscher’s theory to establish a baseline for NGNs to facilitate their integration into the workplace [15]. According to Murray, Sundin, and Cope [15], Duchscher’s transition shock theory and Benner’s theory both start from novice to expert, providing the framework for understanding NGNs transition to practise. Duchscher’s transition shock theory describes the path of new nurses entering the profession, and Benner’s theory identifies delineated competency in five stages: novice, advanced beginner, competent, proficient, and expert. Previous studies have explored the experiences of NGNs during their initial transition to the clinical setting. Rush et al. [11] found that NGNs commonly face challenges related to a lack of confidence, difficulties in managing patient workload, and a lack of support from their peers [11]. Similarly, Liang et al. [3] and Labrague and McEnroe-Petitte [10] identified struggles among NGNs in establishing professional relationships, acquiring necessary knowledge, and developing assertiveness skills [3,10]. The nursing profession is important, as nurses are responsible for taking care of human lives. Moreover, it is crucial to pay attention to NGNs until they become professionals. However, to the best of our knowledge, there has been no research on NGNs experiences in Taif City. Therefore, the objective of this qualitative study is to examine deeply the experiences of NGNs during their first year of practise and investigate the factors that contribute to their successful integration into the workplace and its related difficulties.

## 2. Materials and Methods

### 2.1. Study Design

This study follows a qualitative descriptive design that employs in-depth interviews to explore the experiences of newly graduated nurses during their first year of practise. Qualitative research is an appropriate method for exploring individual experiences and perceptions surrounding a phenomenon, as it allows for the collection of data that goes beyond descriptive statistics [16].

### 2.2. Study Setting

This study was conducted in hospital settings, including governmental, private, and military hospitals within the city of Taif, Saudi Arabia. This study targeted participants who had graduated from an accredited nursing programme and were in their first year of practise.

### 2.3. Study Population

The study population consists of newly graduated nurses (NGNs) who are in their first year of practise within hospital settings in Taif, Saudi Arabia. The inclusion Criteria for this study were newly graduated nurses who must be NGNs who have graduated from an accredited nursing programme and are working full-time in hospitals. NGNs with fewer than 12 months of experience in clinical practise are eligible for inclusion in the study.

### 2.4. Sampling and Sample Size

A purposive sampling technique was used to select NGNs who were interested in participating in the study. This technique involved approaching potential participants through nurse managers in hospitals and explaining the purpose of the study. Therefore, all newly graduated nurses who are in their first year of practise within hospital settings in Taif, Saudi Arabia, were invited. Once contacted, eligible NGNs who expressed an interest in the study were provided with detailed information about the study objectives, procedures, and data confidentiality, and informed consent was obtained before their participation in the study. The sample size for this study was determined through a process of data saturation. This process involves analysing the transcripts of the interviews as they come in and then continuing to collect data until no new information is revealed [17]. According to previous literature, 15–20 participants would be sufficient to achieve data saturation [16]. Therefore, this study’s data saturation was reached throughout the 15 semi-structured interviews.

### 2.5. Data Collection and Analysis

Semi-structured interviews were chosen for data collection to explore personal experiences and thoughts regarding a particular phenomenon. The focus was on obtaining detailed and comprehensive descriptions of the experiences of these nurses as well as examining the factors contributing to the integration of new nurses into the workplace. The interviews included questions such as: “What type of work issues have you faced during your first year as an RN, and why?”, “What makes you more confident?”, and “What do you think would help newly-hired staff nurses in the first year?”.

The data analysis for this study was conducted based on the aims and objectives of the study. The data collected through the interviews was analysed using thematic analysis, which involves identifying patterns, concepts, and themes within the data [18]. The analysis focused on the difficulties that NGNs experience during their first year of practise and the factors that enable them to integrate into their workplace. The thematic analysis process was performed with multiple readings of the interview transcripts to become familiar with the data. The transcripts were re-read to identify the initial codes, which would be combined into candidate themes. These themes were then reviewed, refined, and clustered to identify the final themes and sub-themes. The themes that emerged were compared with the current literature to support their identification and better understand their relevance and impact. The data analysis helped in the identification of factors influencing the integration of NGNs into their workplace and overcoming the challenges that arise during their first year of practise.

## 3. Results

Fifteen newly graduated nurses from Taif City participated in this study: nine female nurses and six male nurses. Their ages ranged between 23 and 25 years old, as shown in Table 1. This study identified two main themes. First, factors contributing to the integration of new nurses into the workplace include three subthemes: the positive role of trainers in a work environment, the gradual handling of patients, pre-employment training, and volunteering. The second theme is difficulties faced by new nurses in a work environment, which includes three subthemes: difficulty dealing with the health system and devices in a work environment; fear of dealing with new patients; and difficulty applying policies and procedures in the workplace.

### 3.1. Factors Contributing to the Integration of New Nurses into the Workplace

#### 3.1.1. The Positive Role of Trainers at the Workplace

The involvement of trainers at the start of the participants’ employment has helped them become familiar with the work environment. Thirteen of the participants agreed that trainers teach them how to deal with and cooperate with patients in different cases, teach them important knowledge and skills, and have answers to all of their questions. Therefore, it had a positive impact and helped them adapt easily, as quoted below:

“I had no idea that I was going to be employed at a critical care unit. I was employed and had a training supervisor who introduced me to the department. In the first week, I took care of zero patients. Afterwards, I was able to take care of stable patients; she would explain things and stay by my side for all of the procedures and on the system when I was entering data. Then I started to only go back to her for things I was not sure about. Her support for me was useful; she stayed for 2 months, and after that, I would only ask her if I was unsure. Gradually, I took care of patients from stable to critical ones.”(Participant 1)

“I had a supervisor, and I will never forget their blessing after God—they guided me for 3 months and taught me everything in all situations—how to deal with patients, how to finish my papers, and how to manage my time during work. They never had any relevant information with all of the nursing procedures and never faced difficulties, even though she was a foreigner. Our language was good and focused throughout the three months; her presence really helped, and we learned quickly and got familiar with everything. She helped us to an extent where we did not feel pressure, and they did not punish us for paper mistakes or any mistakes that did not involve patients; they also encouraged us.”(Participant 7)

However, not everyone received on-the-job training at the beginning of their employment. This was due to either a shortage in the nursing staff or the trainer being on annual leave, as mentioned below:

“In my case, there was no trainer—it was a holiday, and when she came back from her holiday, they placed her in the quality department. In the first month, I taught myself, and from the first week on, they gave me patience. I had early shock, job burnout, and fear at night. Then I learned and asked some of the nurses to help; some of them apologised or refused. After the first month, the trainer came and showed me what I needed, and then new nurses came a month later, and the trainer trained them. At that time, I was holding three new critical cases.”(Participant 15)

#### 3.1.2. Gradual Handling of Patients

At the beginning of their work in the field of nursing, twelve participants gradually provided care to patients. They started by providing nursing care for stable cases. Furthermore, they took care of critical cases gradually until the number of patients increased to the normal range. Furthermore, there were others who advised and supported nurses, such as nursing directors and supervisors. Hence, this contributed to their adaptation to work and the development of their knowledge and skills, as quoted below:

“When I joined the hospital, The first patients I received in the department were chronic, stable cases. After two weeks, I gradually began to catch critical cases—after a month, I caught three critical cases in my shift. However, even if you are an expert, you sometimes need to ask others to help or consult on certain cases.”(Participant 2)

“We are in the critical care unit; we are assigned to two patients according to the type of healthcare condition. It will not deteriorate or be critical; it is an intermediate case. Hence, if I need help, I ask for it. As you know, the cases here are all critical; of course, there are no stable or simple cases here—they all need a lot of procedures, compound medicines, and IV fluids.”(Participant 10)

“Now that I have completed almost 4 months, I can work alone with critical patients and with one stable patient, like the rest of the nurses; praise be to God for help from my colleagues, that is, when I want someone to help me with the child to hold them at any time to implement nursing procedures.”(Participant 14)

#### 3.1.3. The Benefit of Pre-Employment Training and Volunteering

Eight participants indicated that working as a locum in nursing and as a volunteer before working as staff helps with acquiring experience and gaining skills. Thus, some of the new employees do not suffer a lot at the start of their careers, as quoted below:

“I worked as a locum before starting my official work—my desire to work in critical departments—and as you know, personality also makes a difference, whether he or she is self-confident or not—I feel that I can be creative and enjoy my work in an intensive care unit. It will be good when nurses are assigned one patient in the ICU and not the other way around, where one nurse is responsible for three to four cases, because this is not intensive care.”(Participant 4)

“Locum and volunteering increase self-confidence. I experienced this myself. Helping others to nurse increases my efficiency. I had a 12-hour shift because the staff shortages were annoying for me, and there was also an intellectual difference between doctors and nurses. The nurses are serving more than their capacity in the department, and this causes us pressure—permanent cooperation in the department and nursing; every day I learn something new—my schedule and my rest day were as I asked, and they help me if I get new admissions or difficult situations.”(Participant 2)

### 3.2. Difficulties Faced by New Nurses in a Work Environment

#### 3.2.1. Difficulty Dealing with the Health System and Devices in a Work Environment

Ten of the participants indicated that hospitals have electronic systems and a number of different medical devices. They have sometimes not used or seen these devices before or during their studies. They do not have the knowledge or skills to deal with it. Therefore, it is necessary to have a supervisor or mentor teach them how to prepare it and use it in a good and safe way, as quoted below:

“Some of the devices I was trained in, but the respirator was difficult for me. It is difficult to use devices without this information because you may transfer infections or harm patients. I was observing the respiratory and nursing staff, and I was assigned to cases, but I did not understand how to change the settings of the device, so I asked the supervisor many times until I learned. The problem is that most of the devices are not used daily, so it is easy to forget. The good thing is that some workshops from the medical devices department help us.”(Participant 1)

“That is an issue; it was difficult because it is a new environment and there are procedures related to the health and safety of the patient; for example, how do we get the results of analyses and how do we transfer patient information into the system? These are the problems we face, and there are things in the system that if we make mistakes and mistakes are not modified, that is why supervisors are with us step by step so we can learn.”(Participant 7)

“Some devices were new to me, and the trainer explained them to me. It took me about three weeks to use them in a good and safe way—technology has made many things easier, even if they started off difficult, but as we know, technologies help us to finish our tasks more easily.”(Participant 2)

#### 3.2.2. Fear of Dealing with New Patients

Nine participants mentioned that fear sometimes controls them while performing work. This fear can be seen when there are new and unstable disease cases that they have not dealt with before, such as COVID cases. Furthermore, in cases where there is a procedure that has not been performed before or in a different patient, such as a small child or older patient, it is difficult to deal with this during nursing procedures, as quoted below:

“I had no problems when I performed many procedures in nursing; however, difficulties are found with new diseases, new devices, or new regulations. When I start new nursing procedures with patients, it is difficult. I feel fear, but with practise, it becomes easy and can be done without fear.”(Participant 8)

“The worst situation was with a patient who mistreated me because I was new. He was not confident in me. I tried to give him an IV injection, and my hand was shaking because I was afraid. He started to get angry with me. I had only been given an IV once, but the patient was a little fat and needed experienced staff. A second bad situation was when the doctor gave a verbal order, and I was new. I did not know the policy or whether I was supposed to write it down and document it in the patient file. We carried out the doctor’s orders. The problems have been when I sometimes make mistakes. They do not consider that I am new and that I need help!”(Participant 13)

“There were no difficulties, praise be to God, but it was a matter of adapting to the new situation and taking responsibility for work and cases—organising time is important and working with team spirit, especially in critical departments; no one can do without anyone—forming relationships with the staff of my department. We depend on each other, and we share the experience. The problem was when patients died; I used to cry about that until I got used to the situation. I mean, when I had a new patient, I was afraid that I would lose him immediately.”(Participant 12)

#### 3.2.3. Difficulty Applying Policies and Procedures in the Workplace

Health policies and procedures are important. Ten of the participants mentioned this. They also mentioned that all new nurses at work are required to not only read them but also apply them in order to practise them, because most of the problems are due to a lack of practise of nursing policies and procedures during work. This is for a variety of reasons, such as a lack of time and needing to practise, as quoted below:

“Workshops are supposed to be held on infection control and what should go into the yellow box—we need to review before practising because the one in the university differs from that in the hospital. Here, they just showed me the file of policies and procedures, but there was no motivation to read. I took a course on similar medicines under a good trainer. Hence, we need to read it, but we have no motivation or time to do that.”(Participant 1)

“Many things in the hospital system—policies and procedures—are supposed to be explained, not only shown to new nurses, but explained and practised according to priority. The daily procedure is important and different from procedures that happen every year. We need to give the new nurses time to understand because learning abilities differ from one nurse to another. Learning the documentation is more important than nursing needs for any procedure they undertake because it is a daily process. Preparing medicines is also important for them to learn and be intensive in the first period of the appointment.”(Participant 4)

“It is possible to explain the policies and procedures and apply them in practise, much better than handing them to us, so that we are more familiar with medications, for example. Practise is better than most things like sending emails or handling papers.”(Participant 14)

## 4. Discussion

The present study aimed to explore the experiences of newly graduated nurses during their first year of practise, focusing on the factors contributing to their integration into the workplace and the difficulties they faced.

Factors contributing to the integration of new nurses into the workplace emerged as a significant theme in our study. Within this theme, three subthemes were identified: the positive role of trainers in a work environment, the gradual handling of patients, and the benefit of pre-employment training and volunteering. These findings resonate with several previous studies that have explored the transition experiences of newly graduated nurses [1,3,5,6,8,19,20,21,22].

In their qualitative study on the transition challenges of fresh nursing graduates, it was found that the presence of supportive mentors and preceptorship programmes facilitated the integration process [7]. This finding aligns with our subtheme on the positive role of trainers in the work environment, with participants highlighting the importance of having experienced supervisors who guided them, answered their questions, and provided continuous support. Similar to another qualitative study by Gellerstedt et al. [8], which examined the experiences of newly graduated nurses in a trainee programme, our participants emphasised the value of pre-employment training and previous volunteering experiences in enhancing their confidence and skills [8].

The second theme identified in our study was the difficulties faced by new nurses in a work environment. This theme included three subthemes: difficulty dealing with the health system and devices in a work environment; fear of dealing with new patients; and difficulty applying policies and procedures in the workplace. These findings are consistent with previous research, which highlighted the challenges encountered by newly graduated nurses during their transition period [3,4,7]. Furthermore, the lack of awareness of work policies is one of the reasons for conflict between nurses in general [23].

In their integrative review of new graduate nurses’ transition to acute care, Hawkins et al. [24] identified similar difficulties related to the health system and devices. They found that new nurses often struggled with the use of electronic systems and unfamiliar medical devices. Similarly, our participants expressed concerns about lacking knowledge and skills in dealing with certain devices, emphasising the need for proper training and mentorship.

Fear of dealing with new patients was another subtheme that emerged in our study. This finding is in line with the results of Ke and Stocker [4], who explored the processes of growth among new nurses in the workplace. They found that new nurses often experienced anxiety and fear when faced with unfamiliar and challenging patient cases. Our participants shared similar experiences, highlighting the initial fear and apprehension they felt when encountering new and critical cases. However, with practise and support from colleagues, they gradually overcame their fears and gained confidence.

The third subtheme, difficulty applying policies and procedures in the workplace, corresponds to the findings of a study by Lee and Sim [22], which examined the gap between college education and clinical practise for newly graduated nurses. They found that new nurses often struggled to apply theoretical knowledge in real-life settings, particularly when it came to adhering to policies and procedures. Similarly, our participants mentioned the challenges of understanding and applying policies and procedures in their workplace. Therefore, many studies have highlighted the need for more practical training and guidance to bridge this gap effectively [22,23].

Overall, our study’s findings are consistent with the existing literature on the experiences of newly graduated nurses during their transition into the workplace. The themes and subthemes identified in our analysis align with the challenges and facilitators highlighted in the previous literature. These consistencies highlight the significance of addressing these influencing factors and challenges experienced by NGNs.

The study’s strengths include the fact that it utilises a qualitative descriptive design with in-depth interviews to explore the experiences of newly graduated nurses in their first year of practise. Data collection involved semi-structured interviews to obtain detailed descriptions and experiences. Thematic analysis was employed to identify patterns and themes in the data, while ethical considerations and participant confidentiality were ensured. The study provides rich and comprehensive data but is limited by its context-specific nature, potential biases, and small sample size.

## 5. Conclusions

In conclusion, this study sheds light on the challenges faced by NGNs in Taif City and the factors that contribute to their integration into practise settings. The study highlights the importance of support and resources for these nurses as they navigate their early professional years. The findings contribute to the existing literature on nursing practise and offer recommendations for improving the transition process for NGNs. However, it is important to note that the study has certain limitations, such as its specific focus on a particular context and the small sample size. Future research is highly recommended to replicate the study in different settings and with a larger sample size to ensure generalisability across the Kingdom and strengthen the validity of the findings. The result of this study indicates that educational departments in hospitals are likely to support the new nurses (e.g., teaching new nurses to understand and adapt policies to enhance their quality of nursing care). Therefore, it is essential for healthcare policymakers to explain the importance of hospitals policies on quality care.

## Figures and Tables

**Table 1 healthcare-11-02048-t001:** The participants’ characteristics.

Participants	Age	Gender	Experience	Qualification
1	25	F	9 Months	Bachelor
2	25	F	1 Year	Bachelor
3	24	F	7 Months	Bachelor
4	25	F	1 Year	Bachelor
5	24	M	8 Months	Bachelor
6	24	M	5 Months	Bachelor
7	24	M	1 Year	Bachelor
8	25	M	6 Months	Bachelor
9	26	M	1 Year	Bachelor
10	24	M	5 Months	Bachelor
11	23	F	5 Months	Bachelor
12	24	F	1 Year	Bachelor
13	24	F	1 Year	Bachelor
14	24	F	4 Months	Bachelor
15	24	F	1 Year	Bachelor

## Data Availability

Data can be requested from authors.

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
