# Peer review of "The Experiences of Newly Graduated Nurses during Their First Year of Practice"

_healthcare, 2023, doi:10.3390/healthcare11142048_

Round 1

Reviewer 1 Report

I would like to thank the author for the effort to look into the matter of healthcare framework of Saudi Arabia – and experiences of nurses during immediately after their graduation and during the first year of practice. Re the size of the sample, please note that qualitative research is designed to describe a phenomenon, not to validate it, so the sample may be small, as it is in the current draft (and it is perfectly valid as such).

Overall: describing the experience of a particular group in a range of medical facilities is a valid and good research objective. However, the research findings need to be strengthened to provide new insights, as currently they largely confirm other findings on the subject matter. Furthermore, the themes highlighted may be implicit in advance (uncertainty, fear, anxiety, benefits of on the job training, etc.), which make qualitative research shallow. I really see the benefit in expanding on the available findings and strengthening the analytical component of the draft. To this end, please find a few specific suggestions and observations below.

Article Title: the title captures the essence of the research matter. A suggestion is to add geographical area - city of Taif, Saudi Arabia.   

Abstract:

24: a suggestion to either list key findings from conclusions, or remove the reference form the abstract altogether, as currently it does not add to the abstract, as it is formulated.

25: perhaps specify which type of support and what type of resources?

Introduction:

34: a suggestion to remove the word “challenges” as it is overused in policy outputs, and its meaning is too broad to convey a message about the subject matter. 35-36 effectively say the same thing, but in a more specific and meaningful manner. May be worth merging 2 sentences. 38-39 say the same thing.

Sample size: “a sample size of 15-20 NGNs” should be specific, mentioning the number of interviews conducted, not a range.

Discussion: 311-312 – this was stated earlier.

Conclusion: a suggestion to mention the specific geographic location.

Also, 2.2. section earlier justifies the choice of the city of Taif, Saudi Arabia, due to its variation in healthcare models and the heterogeneity of its patient 82 population, including governmental, private and military hospitals. This variety merits a look at whether there are differences across experiences of the nurses at each of these medical facilities. This is work exploring within the paper. Currently, while stated as a justification, the findings are not presented.

I would suggest to merge repitative sentences into one, when they are placed in the same section, and to remove repitative statements across the text. 

Author Response

Article Title

The title captures the essence of the research matter. A suggestion is to add geographical area - city of Taif, Saudi Arabia.   

Done

Abstract:

25: perhaps specify which type of support and what type of resources?

 Done

Introduction:

 34: a suggestion to remove the word “challenges” as it is overused in policy outputs, and its meaning is too broad to convey a message about the subject matter. 35-36 effectively say the same thing, but in a more specific and meaningful manner. May be worth merging 2 sentences. 38-39 say the same thing.

 Done

Methods

Sample size: “a sample size of 15-20 NGNs” should be specific, mentioning the number of interviews conducted, not a range.

 Done

Discussion:

311-312 – this was stated earlier.

 Done

Conclusion:

 A suggestion to mention the specific geographic location.

 Done

Also, 2.2. section earlier justifies the choice of the city of Taif, Saudi Arabia, due to its variation in healthcare models and the heterogeneity of its patient 82 population, including governmental, private and military hospitals. This variety merits a look at whether there are differences across experiences of the nurses at each of these medical facilities. This is work exploring within the paper. Currently, while stated as a justification, the findings are not presented.

Done

English Language:

I would suggest to merge repitative sentences into one, when they are placed in the same section, and to remove repitative statements across the text. 

Done

Reviewer 2 Report

I consider that this work can be accepted for publication if the authors meet the following points:

1. The last paragraph of the introduction mentions the research gap you are addressing with your work; however, you should try harder to make it clearer for readers.

2. In section 2.4, the authors should better describe the characteristics the subjects had to possess to be selected for the study.

3. I believe that the authors should perhaps add in their conclusions the practical implications that their findings should have, both for the educational institutions from which these new nurses graduate and for the companies that hire them, that is, how these institutions can define policies or including procedures to facilitate the insertion of these new graduates.

4. In addition, the authors should try to identify why they consider that the problems of recent nursing graduates upon entering the world of work would be different from those of any other recent graduate, from an engineering program, for example.

5. Increase the number of references from your study.

Author Response

Response to reviewers' comments

Reviewer # 2

Reviewer # 2

Response

1. The last paragraph of the introduction mentions the research gap you are addressing with your work; however, you should try harder to make it clearer for readers.

Done

2. In section 2.4, the authors should better describe the characteristics the subjects had to possess to be selected for the study.

Done

3. I believe that the authors should perhaps add in their conclusions the practical implications that their findings should have, both for the educational institutions from which these new nurses graduate and for the companies that hire them, that is, how these institutions can define policies or including procedures to facilitate the insertion of these new graduates.

Done

4. In addition, the authors should try to identify why they consider that the problems of recent nursing graduates upon entering the world of work would be different from those of any other recent graduate, from an engineering program, for example.

Done

5. Increase the number of references from your study.

2 references were added

Reviewer 3 Report

General considerations:

We believe the paper is within the scope of the journal, it brings empirical data on professional integration of nursing graduates.

The writing is adequate, the title, abstract, and main sections of the paper are consistent with data presented. The aims are clearly stated and the method appropriate. The conclusions are supported by the data presented.

Nevertheless, we have some minor suggestions that we believe the authors should address, before this paper is published, identified below with reference to the paper sections.

Abstract: I think there’s no need to write the words “Methods:”, “Results:” and “Conclusions:” the reader understands what you are writing about.

Introduction

Page 2, line, when referring to “Benner's model and Duchscher's theory” you should provide a clear, descriptive paragraph on this model, not every reader will be an expert on you field.

2.3 Study population

Page 2-3, I don’t think that the exclusion criteria is needed, it seems redundant after presenting the inclusion criteria. But it would be informative to know how many nurses fits your criteria in this hospital.

3. Results

Page 3, line 130-131 and table 1, the sample and characterization is not part of the results, it should be presented earlier when you describe your sapling procedures.

Throughout this section, when you present the results, you mention that “most participants” or “several participants” and “a number of participants”, this should be quantified, you should explicit say how many participants you are referring to in each case.

 In our view, after these small corrections, this paper is ready for publishing.

Author Response

Response to reviewers' comments

Reviewer # 3

Reviewer # 3

Response

Abstract:

I think there’s no need to write the words “Methods:”, “Results:” and “Conclusions:” the reader understands what you are writing about.

 Done

Introduction:

Page 2, line, when referring to “Benner's model and Duchscher's theory” you should provide a clear, descriptive paragraph on this model, not every reader will be an expert on you field.

 Done

2.3 Study population

Page 2-3, I don’t think that the exclusion criteria is needed, it seems redundant after presenting the inclusion criteria. But it would be informative to know how many nurses fits your criteria in this hospital. 

 Done

Results:

 Page 3, line 130-131 and table 1, the sample and characterization is not part of the results, it should be presented earlier when you describe your sapling procedures.

We appreciate your comment, but we apologize for not moving it to another place because it is common appear in results section

Throughout this section, when you present the results, you mention that “most participants” or “several participants” and “a number of participants”, this should be quantified, you should explicit say how many participants you are referring to in each case. 

 Done

In our view, after these small corrections, this paper is ready for publishing.

We appreciate that

Round 2

Reviewer 2 Report

My comments have been satisfactorily addressed.